# Understanding the Pivotal Role of the Vagus Nerve in Health from Pandemics

**DOI:** 10.3390/bioengineering9080352

**Published:** 2022-07-29

**Authors:** Claire-Marie Rangon, Adam Niezgoda

**Affiliations:** 1Child Neurologist and Pain Specialist, INWE’CARE Medical Center, 92210 Saint-Cloud, France; 2Chair and Department of Neurology, University of Medical Sciences, 60-355 Poznań, Poland; adamniezgoda@wp.pl

**Keywords:** pandemics, COVID-19, HIV, prion, non-invasive vagus nerve stimulation, neuroimmunometabolism

## Abstract

The COVID-19 pandemic seems endless with the regular emergence of new variants. Is the SARS-CoV-2 virus particularly evasive to the immune system, or is it merely disrupting communication between the body and the brain, thus pre-empting homeostasis? Retrospective analysis of the COVID-19 and AIDS pandemics, as well as prion disease, emphasizes the pivotal but little-known role of the 10th cranial nerve in health. Considering neuroimmunometabolism from the point of view of the vagus nerve, non-invasive bioengineering solutions aiming at monitoring and stimulating the vagal tone are subsequently discussed as the next optimal and global preventive treatments, far beyond pandemics.

## 1. Introduction: Searching for a Single Anti-Infectious Solution

In the context of globalization, it is crucial to rapidly protect the world population against new pathogens. Despite major advances in medical research, the last pandemic has underscored the limits of our societies in overcoming atypical viruses, compelling us to use ancestral methods such as lockdowns.

In the race against time during pandemics, two main strategies usually prevail [1]: drug repurposing, and the development of novel therapies, including vaccines as a longer-term solution. Both strategies are time-consuming, although unequally, because clinical trial assessment is an unavoidably long process. Moreover, these strategies have to be repeated for each specific pathogen involved—potentially even several times during the same pandemic—due to the emergence of variants. Unfortunately, pandemic outbreaks are meant to intensify over the years because of identified risk factors that are more or less susceptible to human action, from the illegal wildlife trade to climate change [2]. In this regard, the prospect of a unique, ready-to-use, therapeutic solution would be unhoped but realistic, provided that the pathophysiology of the host defense is considered first and foremost.

Effector-triggered immunity (ETI) is a conserved mechanism of pathogen sensing, based on the detection of combinations of critical host molecules signaling the presence of pathogens, subsequently activating innate immune signaling pathways and, ultimately, driving inflammatory responses (see [3] for a review). Nevertheless, although cytokine production is necessary to protect against pathogens, excessive systemic inflammation can promote organ failure and death. Among others, neural mechanisms are deeply involved in the control of immune responses at every level (i.e., local, systemic, and central levels) [4]. Autonomic regulation is of particular interest because it allows a bidirectional reflex regulation of systemic inflammation [5] thanks to the vagus nerve—the longest cranial nerve—and its main neurotransmitter acetylcholine, most notably through a mechanism called the “Cholinergic Anti-inflammatory Pathway (CAP)” [6]. Therefore, taking control over vagus-nerve-mediated homeostasis should turn out to be a single solution against all pathogens.

This article aims at questioning the crucial role of the vagus nerve in pathogenic invasion and neuroimmunometabolic dysfunction leading to excessive inflammation, from the analysis of the last two (and ongoing) pandemics—namely, COVID-19 and AIDS—in order to upgrade future therapeutic strategies worldwide.

## 2. Invasion of the Vagus Nerve by Pathogens Appears as a Common Key Step in Host Defense in the Last Two Pandemics

Interestingly, SARS-CoV-2 and HIV-1 do share similarities [7,8]. Indeed, despite engaging different entry receptors, target cells, and transcription and downstream processes, HIV-1 and SARS-CoV-2 follow similar principles of class-I-glycoprotein-mediated viral fusion and entry. The glycoproteins of both SARS-CoV-2 and HIV-1, named spike and Env, respectively, are composed of an N-terminal attachment domain (S1 and gp120, respectively) mediating receptor binding, and a C-terminal fusion domain. Thanks to this glycoprotein similarity, shared with neurotoxins, HIV-1 and SARS-CoV-2 can bind to nicotinic acetylcholine receptors (nAChRs) [9,10,11,12,13]. Thus, considering the various subunit assemblies of this pentameric acetylcholine receptor [14] for a review, nAChRs have been hypothesized to be co-receptors for SARS-CoV-2 [9] as well as for HIV-1 [15,16,17].

The nicotinic acetylcholine receptors (nAChRs) are widely distributed throughout the nervous and immune systems (see [18] for a review), especially in cholinergic neurons. The vagus nerve endings (parasympathetic 10th cranial nerve) largely contribute to the cholinergic control of inflammation [5,6]. nAChRs are expressed in the nodose ganglia of rodents [19], the latter gathering the cell bodies of vagal afferents. 

As the N-terminal binding subdomain of many viruses and of neurotoxins facilitates internalization of the virus [20], after binding to nAChRs, SARS-CoV-2 and HIV-1 could enter the vagus nerve, wherein they induce vagal dysfunctions [21,22,23,24,25,26,27] before finally invading the brain [28,29]. As a matter of fact, the vagus nerve is a common way for various pathogens to invade the brain [30]. For instance, in prion disease, PrP^C^ was shown to binds to nAChRs [31] to invade the vagus nerve [32] and, subsequently, the brain. 

As SARS-CoV-2 has 2–3 times more spike proteins per virus than HIV-1 [8], and as the vagally supplied entrance organs are more numerous in COVID-19 than in AIDS (nose, mouth, lungs, and guts versus urogenital tract, respectively), the invasion of the vagus nerve by SARS-CoV-2 is likely to be more extensive in COVID-19. Adverse outcomes (e.g., death) may not come from the infection itself, but may depend on the importance of vagus nerve invasion and subsequent incapacitation to restore homeostasis. Indeed, the decreased vagal tone, reflected by heart rate variability (HRV) measures—one of the best prognosis factors in COVID-19 [33,34,35]—has also been correlated with HIV outcomes [36,37] and, more broadly, with the outcomes of bacterial and fungal infections in mice [38]. Moreover, the epigenetic marker miR 146a-5p has also been correlated with both COVID-19 prognosis [39,40,41,42] and HIV progression [43,44]. Interestingly, miR 146a-5p is also considered to be a unique biomarker of virus- and prion-induced inflammatory neurodegeneration [45]. All of these arguments support the idea that the prognosis of both COVID-19 and AIDS depends on the ultimate damage to the brain [46], after vagus nerve invasion and impairment by various pathogens.

## 3. The Vagus Nerve Seems Essential to Neuroimmunometabolism and Health

In fact, the underlying rationale for the pivotal role of the vagus nerve in maintaining health concerns its role in pathogens’ invasion of the host. As supported by anatomical and physiological bases, the vagus nerve has the potential to play a major role in neuroimmunometabolism—the emerging interface between immunometabolic regulation of the nervous system and neuroinflammation [47].

The vagus nerve can allow direct regulation of energy supply to the brain, because it anatomically connects the gut (i.e., the site of the uptake of nutrients) and the digestive organs to the brainstem [48]. Thus, focused ultrasound stimulation of three distinct vagus-nerve-supplied organs (i.e., the liver, pancreas, and intestine) was able to prevent hyperglycemia following endotoxin exposure in an experimental model [49]. Moreover, among endocrine and immune communication pathways, the vagus nerve appears as the “fastest and most direct way for the microbiota to influence the brain” (see [50] for a review). 

Remarkably, the constant bidirectional communication within the microbiota–gut–brain axis, mediated by the vagus nerve, is essential for maintaining homeostasis [51]. Indeed, several neuronal populations controlling feeding, showing unique transcriptional and chromatin accessibility landscapes, were recently identified in the brainstem’s dorsal vagal complex (DVC) [52]. In addition, the vagus nerve coordinates the peripheral (via the cholinergic anti-inflammatory pathway) and central (via the hypothalamic–pituitary axis) stress responses, thereby modulating neuroimmune responses globally to restore homeostasis [53]. 

As a consequence, vagus nerve stimulation is a powerful anti-inflammatory tool (see [54] for a recent review), provided the metabolic needs are met. Today, the anti-inflammatory properties of drugs are even assessed according to their vagus-nerve-dependent effects [55]. Nevertheless, upon infection, the heightened activity of the immune system requires an increased rate of metabolism and energy expenditure. In case of insufficient disposable energy, excessive and damaging host inflammation occurs, leading to cytokine storm or acute respiratory distress syndrome (ARDS)—both seen in critical COVID-19 patients [56]. Indeed, some cases of ARDS (non-mediated by SARS-CoV-2 infection) were shown to be controlled by n-3 fatty acids, possibly through their metabolism to specialized pro-resolving mediators (SPMs), in both experimental models and clinical trials [56]. Interestingly, this new superfamily of lipidic mediators (SPMs) involved in the resolution of inflammation (including the lipoxins, resolvins, protectins, and maresins) was recently shown to be enhanced in vitro by human vagus nerve electrical stimulation [57]. Likewise, long-COVID patients present with persistent inflammation, decreased glucose metabolism, and autonomic dysregulation involving the vagus nerve [58], reminiscent of chronic inflammation [26], vagal dysfunction [26], and the changes in glucose metabolic status of T cells and monocytes [59,60] in HIV-infected individuals.

The crosstalk between glucose metabolism status and the anti-inflammatory power of the vagus nerve could be mediated by neuropeptides [61]—in particular by vasoactive intestinal peptide (VIP), along with its receptors and downstream pathway (see [62] for a recent review). Indeed, VIP is released upon vagus nerve stimulation (VNS) [63], is synthesized in the gut, the brain, and the vagus nerve [64], and can be co-transmitted with acetylcholine—most notably in exocrine glands [65]. Moreover, VIP is considered to be a key regulator of the energy metabolism of glia [66], a secretagogue in the pituitary [67] and adrenal medullae [68], and a T-lymphocyte immunoregulator [69].

Lastly, vagus-nerve-mediated regulation of neuroimmunometabolism seems to make use of epigenetic mechanisms—most notably of microRNAs (miRNAs) [70]. unsurprisingly, the much-vaunted miR146a-5p, correlated with HRV and COVID-19 outcomes, has also been involved in metabolic memory [71]. The last generation of optogenetic tools [72] are likely to give valuable insights into the vagus nerve’s physiology, allowing us to finally understand the characteristics of vagus nerve signals transferred throughout the body. The differences in targeting either the vagal efferent or afferent fibers need to be precise [73], as along with selective vagus neuromodulation [74]. Then, new preventive therapeutic strategies should target global neuroimmunometabolic homeostasis, ideally through genetically guided manipulation of select pools of nerve fibers [48] and minimally invasive vagus nerve stimulation modalities.

## 4. Discussion: Novel Therapies Targeting Vagus Nerve Stimulation for Pandemics

Although various therapeutic approaches have been developed in light of the AIDS and COVID-19 pandemics, remarkably, none (neither antiviral medications nor vaccines) has eradicated infection so far. Conversely, retrograde axonal transport of pathogens—most notably through the vagus nerve—has been suggested for more than 10 years in order to protect against neurotropic viruses [75]. Likewise, the efficiency of VNS against infection has been assessed in various animal models for more than 20 years [76] and was also suggested as a potential adjunct solution against infection in humans 15 years ago [77], as well as for treating depression induced by HIV [78] and COVID-19 [79]. Nevertheless, no clinical trials have ever been launched for anti-infectious purposes until 2020, during the first wave of the COVID-19 pandemic. 

Three different articles [80,81,82], using three different non-invasive VNS techniques, started randomized controlled trials in 2020 in France (SOS-COVID-19) and Spain (SAVIOR I), and in 2021 in Austria, respectively, enrolling stage 3 hospitalized COVID-19 patients requiring oxygen supply. Rangon et al. [80] treated the patients with four semi-permanent needles on the conchae of both outer ears (without concurrent electrical stimulation) to stimulate the auricular branch of the vagus nerve, once within the first 3 days after admission. Tornero el al. [81] used the GammaCore^R^ device to provide two consecutive 2-min doses of cervical non-invasive VNS, three times daily (5 kHz sine wave burst lasting for 1 ms, repeated once every 40 ms for 2 min per stimulation). Seitz et al. [82] used percutaneous auricular VNS via three miniature needle electrodes (Auristim^R^ device) inserted into vagally supplied regions of one auricle. The device delivered intermittent electrical stimulation (3 h on/3 h off) with a 1 Hz frequency, 0.5–0.9 mA peak current (1.5 mA maximum), 1 ms pulse width, and 3.8 V fixed amplitude. No study at all was able to show a significant improvement in the clinical outcomes of the COVID-19-positive inpatients (clinical status assessed either too early (14 [80] and 5 days [81] after the first VNS session), or not at all [82]), but a good tolerance of the neuromodulation treatments was confirmed. Interestingly, two studies reported a significant improvement in the biological inflammation status of the patients (regarding C-reactive protein and procalcitonin levels [81]; or C-reactive protein, TNF alpha, DDIMER, and IL-10 levels [82]).

More clinical trials are definitely needed to validate the efficiency of non-invasive VNS in sepsis, even if data from animal studies are very promising [83], given the complexity of the levels of regulation and the diversity of expression of the cholinergic system. For instance, in addition to the impact of smoking status in humans, CHRFAM7A is a unique human gene (expressed neither in primates nor rodents) that encodes a dominant negative inhibitor of the α7 nicotinic acetylcholine receptor, making it difficult to infer results in humans from animal models’ data [84]. The International Consortium on Neuromodulation for COVID-19 (ICNC, www.covidneuromod.org, accessed on 15 May 2020) was created during spring 2020 in order to support the rapid deployment and clinical validation of neuromodulation technologies—especially VNS devices—through multiple modalities, including electricity (transcutaneous cervical or auricular stimulation), ultrasound (percutaneous needle electrode close to the cervical vagus nerve), and focused ultrasound (spleen, liver) (see [85] for a review). Because of their absence of side effects, non-invasive solutions such as monitoring and restoring the vagal tone should also be systematically assessed in clinical trials during epidemics—for instance, in the recent acute and severe hepatitis of unknown etiology in children, probably involving an adenovirus [86], or in case of a monkeypox epidemic [87].

The era of physiolomics has come [88]. HRV monitoring using commercial wearable devices or apps as sensitive but non-specific health indicators (as proposed during COVID-19 [89,90]) could be followed by a more specific diagnosis through blood or salivary microRNAs [91] and analyzed using machine learning systems in order to track, diagnose, and date infection [92]. For instance, a decrease in miR146a-5p may indicate HIV infection [93] and could serve as both a diagnostic and prognostic factor in COVID-19 [36,37]. Consequently, HRV monitoring is undoubtedly worth including in smartphone-based or other expert systems for disease prediction at early stages, or for outcome prediction, which is not yet performed routinely [94,95,96].

Therefore, medical advances during the COVID-19 pandemic could incidentally improve general health worldwide. Very interestingly, miR146a-5p is also involved in aging, cardiovascular and metabolic diseases, cancers, autoimmune diseases such as rheumatoid arthritis [97] or multiple sclerosis [98], and neurodegenerative diseases [99]. Likewise, HRV is also a prognostic factor in several non-infectious diseases [100]. Using affordable non-invasive vagus nerve monitoring could be helpful for early diagnosis (as recently suggested for type 2 diabetes mellitus [101]).

Concurrently, VNS is one of the therapeutic options in many inflammatory diseases, providing good outcomes in rheumatoid arthritis (RA), for instance. In 2016, Koopman et al. demonstrated that in RA patients, an implantable vagus-nerve-stimulating device significantly inhibited tumor necrosis factor production for up to 84 days and was able to significantly improve the clinical disease severity [102]. VNS has mainly been a last-resort therapeutic option so far, e.g., in refractory epilepsy. Despite that, after two years of invasive VNS, roughly half of patients experienced at least 50% reduced seizure frequency [103]. Moreover, in a recent pilot study, non-invasive transcutaneous cervical VNS was delivered to 36 patients with RA of either high or low disease activity [104]. Even in the 16 patients with high disease activity, non-invasive VNS was able to reduce the disease activity score, C-reactive protein, and interferon gamma levels after 4 days.

As diseases are already or soon to be diagnosed earlier, minimally invasive VNS modalities should also be proposed as a first-line treatment in several indications [54,105]. Remarkably, transcutaneous auricular VNS was recently advocated for the treatment of endometriosis [106], which is especially interesting since the diagnosis of endometriosis has become available with a simple salivary test [107].

As many common (e.g., type 2diabetes mellitus) [108], uncommon (e.g., narcolepsy [109]), and emerging common disorders (e.g., autism [110], long-COVID syndrome [111]) are immune-mediated, complete understanding of the vagus nerve’s role is likely to become an all-in-one “Holy Grail” for the prevention, treatment, and prognosis of a large spectrum of disorders. Notably, chronic fatigue syndrome [112]—still a challenge for physicians—is characterized by debilitating fatigue despite rest without validated biomarkers [113]. Chronic fatigue syndrome has been presumably linked for years to a nonspecific infection of the vagus nerve by pathogens [114], to finally be recognized as the major feature of post-acute sequelae of COVID-19 [111]. A recent double-blind, sham-controlled pilot study assessed the impact of a four-week at-home treatment with self-administered electrical transcutaneous auricular vagus nerve stimulation (Soterix Medical, Inc. (Woodbridge, New Jersey, United States); 25 Hz, 500 μs pulse width, tonically on for 1 h, twice daily, 6 days per week, twice individual perceptual threshold intensity) to manage long-COVID symptoms. This VNS treatment was not only safe and highly compliance-rated, but also revealed interesting trends in reducing mental fatigue scores [115]. Therefore, transcutaneous auricular VNS appears to be a promising therapeutic option to manage chronic fatigue syndrome.

## 5. Conclusions: The Pivotal Role of the Vagus Nerve, beyond Pandemics

The analysis of the last two pandemics underlines the central role of the vagus nerve in maintaining homeostasis, most notably via regulation of neuroimmunometabolism. Many viruses or pathogens invade and hijack the vagus nerve, presumably after co-binding to nicotinic acetylcholine receptors, in order to enter the brain, resulting in brain lesions and poor outcomes. The rapidity and the scale of SARS-CoV-2 infection in particular urge the repurposing of drugs and/or therapeutic solutions.

The efficiency of non-invasive vagus nerve stimulation—either cervical or auricular—definitely ought to be assessed in larger clinical trials involving severely infected patients, in order to be able to react faster in the event of another pandemic or beyond. Indeed, the impact of such clinical trials is also likely to be useful for the prevention and treatment of diseases that have been poorly managed so far. “What doesn’t kill us makes us stronger…”.

## Data Availability

Not applicable.

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
