# Peer review of "Understanding the Pivotal Role of the Vagus Nerve in Health from Pandemics"

_bioengineering, 2022, doi:10.3390/bioengineering9080352_

Round 1

Reviewer 1 Report

I would have liked to see a more robust discussion of how vagus nerve stimulation can modulate disease and inflammation.   There are a number of studies from rheumatoid arthritis to long covid that have been associated with improved outcome with stimulation.  

Author Response

We thank the Reviewer for the comments.

First, we revised the introduction is order to make it clearer.

We also added discussion about VNS efficiency on inflammation, notably by citing studies on Rheumatoid Arthritis and Long COVID symptoms. 

We hope this new version will satisfier the Reviewer.

Sincerely,

CMR and AN

Reviewer 2 Report

This is an interesting narrative review. There is no mention of how the particular references were selected.

Author Response

We thank the Reviewer for the comments.

We revised the introduction and the presentation of the results.

We want to underline that our article is a perspective rather than a narrative review, as acknowledged by the two other reviewers. Indeed, although the structure of Perspectives is similar to the structure of Reviews, Perspectives emphasize future directions of the field based on the personal assessment of the authors.

Therefore, the particular references were selected to answer (and support) the different questionings of the authors (in each paragraph).

We hope that this new version of the article will satisfy you.

Sincerely,

CMR and AN

Reviewer 3 Report

The authors presented their perspective about the "pivotal role of vagus nerve in health" in a very concise way. Overall, this short perspective paper is well written and may attract societal attention onto this topic. This paper would potentially lead to a broad impact and readership. But I would still like to suggest the authors to include more vagus nerve stimulation modalities, such as magnetic/ultrasound/chemical/genetic stimulation approaches that are able to modulate neuroimmunometabolism processes.

Some minor comments:

1. I am confused by the statement of "...subsequently inadequate activation level of ... being exhausted" in lines 100-101. Please rewrite it.

2. Please change "precised" in line 154 to be "precise".

3. Please include the full name of HRV at its first appearance.

4. The sentence in lines 188-189 should be removed.

Author Response

We sincerely thank the Reviewer for the encouraging comments.

We corrected the sentences as requested by the Reviewer.

We also modified the introduction and the discussion in order to lay more emphasis on the different modalities of VNS.

Sincerely,

CMR and AN
